# The Intestinal Barrier and Its Dysfunction in Patients with Metabolic Diseases and Non-Alcoholic Fatty Liver Disease

**DOI:** 10.3390/ijms23020662

**Published:** 2022-01-08

**Authors:** Roberta Forlano, Benjamin H. Mullish, Lauren A. Roberts, Mark R. Thursz, Pinelopi Manousou

**Affiliations:** Division of Digestive Diseases, Department of Metabolism, Digestion and Reproduction, Imperial College London, London W2 1NY, UK; r.forlano@imperial.ac.uk (R.F.); b.mullish@imperial.ac.uk (B.H.M.); lauren.roberts@imperial.ac.uk (L.A.R.); m.thursz@imperial.ac.uk (M.R.T.)

**Keywords:** NAFLD, gut permeability, gut microbiota

## Abstract

Non-alcoholic fatty liver disease (NAFLD) represents an increasing cause of liver disease worldwide, mirroring the epidemics of obesity and metabolic syndrome. As there are still no licensed medications for treating the disease, there is an ongoing effort to elucidate the pathophysiology and to discover new treatment pathways. An increasing body of evidence has demonstrated a crosstalk between the gut and the liver, which plays a crucial role in the development and progression of liver disease. Among other intestinal factors, gut permeability represents an interesting factor at the interface of the gut–liver axis. In this narrative review, we summarise the evidence from human studies showing the association between increased gut permeability and NAFLD, as well as with type-2 diabetes and obesity. We also discuss the manipulation of the gut permeability as a potential therapeutical target in patients with NAFLD.

## 1. Background

Non-alcoholic fatty liver disease (NAFLD) represents a leading cause of liver disease and referrals to liver transplantation worldwide [1]. Histologically, NAFLD encompasses a spectrum of pathological disorders characterised by macro-vescicular fat accumulation (steatosis, NAFL) with or without hepatocellular injury and/or inflammation (non-alcoholic steato-hepatitis (NASH)) and a variable degree of fibrosis through to cirrhosis [2,3]. It has been estimated that the prevalence of NAFLD ranges between 19–46% with higher ranges among those with metabolic factors, i.e., type-2 diabetes mellitus (T2DM) and obesity [4].

The development and progression of fatty liver to NAFLD with fibrosis and/or NASH is now being explained with a multi-hit hypothesis, where a plethora of dietary, environmental, and genetic factors contribute to the disease along with the worsening of insulin resistance. Dietary elements, both in terms of overall calorie intake and specific dietary patterns, may contribute to the development of NAFLD/NASH. Specifically, high-fat diets, increased fructose intake and red meat intake have been associated with worsening hepatic steatosis and the induction of a pro-inflammatory status [5,6,7]. Moreover, there is a large body of work showing that the gut microbiome plays an essential role in disease activity in patients with NAFLD [8]. Overall, there is a bidirectional relationship between the gut and the liver, which is mediated through the portal vein, the biliary tract and the systemic circulation. With the portal vein being the main blood supply, the liver is at the forefront for the metabolism of endogenous as well as exogenous substrates coming from the intestine [9,10]. The interactions between the liver and the gut, the so-called “gut–liver axis”, result from a complex interplay between the gut and the liver, mediated by the immune system, which ranges from immune tolerance to immune activation. Changes in the gut microbiome composition, gut permeability and the translocation of pro-inflammatory bacterial by-products are now included among the factors involved in the progression of liver disease in this population [11].

Interestingly, despite the increasing incidence of NAFLD worldwide and the endeavours made in drug development, there is no licensed treatment at present. In this sense, understanding the factors involved in the modulation of the gut–liver axis may provide new insights into the pathophysiology of the condition, together with potential therapeutical targets for treating the disease.

In this review, we discuss the role of altered gut permeability in patients with metabolic disease and NAFLD. We also summarise the evidence from human studies supporting the targeting of the intestinal barrier as a potential therapeutical option for treating the disease.

## 2. Definition of Gut Barrier and Gut Permeability

The intestinal barrier function is defined as the ability of the mucosa and of the components of the extracellular barrier to prevent the exchange between the intestinal lumen and the tissues [12]. Conversely, intestinal permeability refers to the property that allows such exchange. The gastro-intestinal (GI) mucosa is a semi-permeable barrier with multiple properties, such as absorption of nutrients and immune sensing. The gut barrier plays an important role in limiting the passage of potentially pathological molecules and microorganisms into the systemic circulation. Moving from the luminal to the basolateral layer, the intestinal barrier includes the gut microbiota, the mucus layer, the monolayer of epithelial cells and then the immune cells located in the lamina propria. The mucus layer represents a physical barrier that separates the microbiota and large molecules from contacting the epithelial cells, but also acts as a facilitator for the passage of small molecules. Ultimately, the mucus layer defends the epithelium from acid and digestive enzymes. In such distribution, the bacteria from the intestinal microbiome are mainly restrained within the outer part of the film of mucus [13] (Figure 1).

The intestinal epithelium represents a crucial component of the intestinal mucosal barrier. The epithelial layer is composed of different cell populations, such as enterocytes, goblet cells, enteroendocrine cells and immune cells. Specifically, the enterocytes, which are the most abundant cells within the epithelium, display a protective function and modulate the uptake of nutrients and other substances from the intestinal lumen. The goblet cells are actively involved in the secretion of mucus, while the enteroendocrine cells produce GI hormones, peptides and neurotransmitters. Finally, the T-cells and monocytes, mainly located in the lamina propria, as well as the Paneth cells, mainly located in the crypts, participate in the innate and acquired immune response to environmental factors [14] (Figure 1).

The tight junctions (TJ) are the main structures forming the complex for cell-to-cell adhesion that polarises the intestinal epithelium, as they regulate the passage of ions, and therefore, create a difference in potential across the tissue. Among other structures, the hemi-desmosomes are also important for the adhesion of the epithelial cells to the lamina propria [15]. Of note, products may cross the epithelium from the lumen using different pathways, which depend mainly on their chemical properties, such as size and hydrophobicity. Small, hydrophilic and lipophilic compounds can use the transcellular route to cross the plasma membrane of the enterocytes. Ions, water and larger hydrophilic compounds between 400 Da and 10–20 kiloDalton (kDa) are transported using the paracellular route between enterocytes, which is principally regulated by TJs. Several other macro-nutrients, such as amino acids, vitamins and carbohydrates may cross the enterocytes actively by using dedicated transporters. Larger peptides, proteins and bacteria may move to the systemic blood stream via a combination of pathways.

## 3. Methods for Assessing the Gut Permeability

Several techniques have been employed to date to measure the intestinal permeability both in vitro and in vivo. Commonly, permeability has been assessed as transepithelial electric resistance (TEER) measured across monolayers of specific cell lines or biopsies of the GI mucosa. The TEER is usually measured in Ohms and it is a quantitative measure of the integrity of the gut barrier. The classic setup used for the measurement of TEER consists of a cellular monolayer cultured on a semi-permeable insert that compartmentalises the model into an apical (upper) and basolateral (or lower) space. Two electrodes are placed in the upper and lower compartment and therefore, they are physically separated by the cellular monolayer. Hypothetically, the ohmic resistance can be calculated by applying a direct current voltage to the electrodes and measuring the actual current between them. The ohmic resistance is calculated based on Ohm’s law as the ratio of the voltage and current. These experiments are usually carried out with the use of an epithelial volt/ohm meter (EVOM) employing an alternating current [16]. Other techniques for measuring the permeability across cell monolayer in vivo have used probe molecules, such as dextran 4 or 40, exploiting a similar concept [17].

Interestingly, the intestinal permeability may also be measured in vivo (Table 1) as the urinary excretion of ingested probes (mainly saccharides) that cross the intestinal epithelium by the paracellular pathway are filtered by the glomerulus and excreted in the urine without active reabsorption. The majority of the saccharides used in these techniques are absorbed in the small bowel and colon, with different timing of urinary excretion reflecting different regional absorption throughout the GI tract [18]. Specifically, the lactulose to mannitol ratio is commonly used to assess the small intestine permeability, as both sugars are degraded by colonic bacteria [19]. Sucralose may provide an estimation of both small and large intestine permeability, as this sugar can be absorbed along both intestinal tracts [20].

Recent studies have also suggested that serum levels of fatty acid-binding proteins (FABP) and zonulin may provide an estimation of gut permeability in humans. Specifically, FABP-2 are small cytosolic proteins that transport fatty acids and include several isotypes that are expressed in different tissues, such as heart, liver, intestine, muscle and adipocyte [21]. Intestinal FABP (i-FABP or FABP-2) is uniquely located in mature small-intestinal enterocytes. Moreover, its peculiar position within the intestinal villi facilitates its leakage into the circulation when damage to the intestinal mucosa occurs [22]. Early studies confirmed that FABP-2 could be detected in plasma or urine as a result of intestinal ischemia in both animals [23] and in humans [24]. In the field of liver disease, individuals with chronic hepatitis B (CHB) and chronic hepatitis C (CHC) have shown higher plasma levels of FABP-2 compared to controls, suggesting some degree of enterocyte death [25]. Similarly, zonulin is involved in the assembly of TJ and therefore in the regulation of mucosal permeability. Previous studies found an increased level of zonulin in patients with intestinal diseases characterised by leakier gut, such as inflammatory bowel disease and coeliac disease [26,27]. Faecal calprotectin has also been suggested as an ex vivo serum marker of increased gut permeability [28,29,30]. More recently, a novel non-invasive assessment of gut permeability was developed based on fluorescence spectroscopy [31].

## 4. Translocation of Intestinal Bacterial Products and Hepatic Inflammation

The translocation of intestinal bacterial products represents the crucial pathological event that follows increased intestinal permeability.

The term intestinal bacterial translocation refers to the passage of viable microorganisms from the gut lumen toward the mesenteric lymphatic system, and subsequently, towards the bloodstream to extraintestinal sites, such as the spleen and liver [32]. Notably, a certain degree of bacteria and bacterial products may translocate even under physiological conditions. However, in normal conditions, the paracellular pathway is limited to molecules up to 20 kDa. As such, bacteria or large particles cannot translocate across the epithelium using this pathway. In a pathological status, the uptake of bacteria occurs mainly via the Peyer’s patches M-cells, and to a lesser extent via the enterocytes [33]. Specifically, the M-cells present a poorly organised border with short and irregular microvilli, which facilitates the internalisation of bacteria, viruses and large molecules from the intestinal lumen to the lymphoid system.

The bacterial translocation is crucial for the modulation of the immune system and the development of immune tolerance. In healthy humans, the liver is exposed to small amounts of bacterial products, and specifically, lipopolysaccharide (LPS), which is the dominant molecule on the surface of Gram-negative bacteria. Elevated LPS levels are often found in the plasma of patients with GI and non-GI inflammatory diseases and has been correlated with bacterial translocation [34,35]. Overall, hepatic inflammation results from the complex interaction of monocytes, resident macrophages (Kupffer cells), neutrophils, parenchymal hepatocytes and liver sinusoidal cells [36,37]. The hepatocytes and the Kupffer cells interact with portal and systemic metabolites, such as pathogen-associated molecular patterns (PAMPs) and damage-associated molecular patterns (DAMPs), and translate them into a cascade of inflammatory events and metabolic dysfunction in the liver [37,38]. Subsequently, the liver transits from an immune-tolerant to an immune-active state, with the further production of inflammatory cytokines, such as transforming growth factor-beta (TGF-β), interleukin-1 (IL-1), interleukin-6 (IL-6) and tumour necrosis factor-alpha (TNF-α).

Another mechanism by which endotoxin may induce liver damage and inflammation is by contributing to an increase in oxidative stress. In a group of patients from the PLINIO study, those with NAFLD presented greater systemic oxidative stress compared to healthy individuals and the levels of oxidative markers increased proportionally with LPS [39]. Interestingly, in this study, compliance to the Mediterranean diet was associated with lower pro-oxidative status. In another recent study, patients with NAFLD showed that higher levels of malondialdehyde (MDA) were associated with LPS, systemic inflammation and hepatic fat content, as measured by magnetic resonance [40]. Ultimately, MDA levels have been associated with the presence of significant fibrosis, as expressed by Fibromax, in men with NAFLD [41].

## 5. Gut Permeability in Patients with T2DM

The presence of T2DM is known to be associated with low-grade systemic inflammation as well as with alterations in the intestinal barrier function. Specifically, gut permeability has been studied extensively in vivo using the Dextran FITC assay in animal models and was increased in diabetic obese mice [42,43]. Interestingly, such changes were modifiable to some extent when the gut microbiome was manipulated, i.e., with the administration of prebiotics [42] or antibiotics [43].

There is also growing evidence supporting the presence of increased gut permeability in patients with T2DM, expressed as increased serum levels of LPS [44,45]. When compared to non-diabetics, patients with T2DM show significantly higher levels of LPS [46]. Moreover, patients with T2DM not only have higher absolute concentrations of LPS, but also higher postprandial excursions of LPS following a high-fat meal [44]. Furthermore, LPS levels were also predictive of developing T2DM in a 10-year follow-up in the FINRISK97 cohort [45]. These results suggest a chronic increase in LPS and also a higher susceptibility to further increase in LPS as a response to diet in patients with T2DM.

Changes in the composition of the gut microbiota have also been described as contributing to an impaired mucosal barrier [47]. Specifically, microbial perturbations have been associated with elevated levels of LPS in the systemic circulation of patients with T2DM [47]. Systemically, the LPS interacts with toll-like receptor 4 (TLR-4) and activates a pro-inflammatory cascade, mediated by the release of cytokines, adhesion molecules and reactive oxygen species [48]. Moreover, in patients with T2DM, decreased levels of glucagon-like peptide 2 (GLP-2) have also been associated with the disruption of zonulin-1, occludin and claudin-1, resulting in abnormalities in the TJ barrier [49].

Patients with T2DM may also exhibit other peculiar mechanisms by which intestinal permeability may be impaired. For instance, in vitro experiments have shown that leptin may modulate the expression of TJ proteins directly [50]. Of note, leptin is a hormone that regulates the appetite sensation; its dysfunction has been described among patients with T2DM [51]. A previous study involving animal models demonstrated that consumption of a high-sugar diet facilitates the degradation of the mucus barrier via the selection of mucus-degrading bacteria [52]. Furthermore, hyperglycaemia has been shown to damage the intestinal epithelial cells directly by altering TJ integrity, with a mechanism that depends on glucose transporter-2 (GLUT2) [53]. In the same study, when hyperglycaemia was corrected, the deletion of inhibition of GLUT-2 was able to restore barrier function [53]. Finally, inadequate glycaemic control has been associated with increased translocation of microbial products in the systemic circulation [53].

## 6. Gut Permeability in Patients with Obesity

It has been previously described that the Western diet may be responsible for changes in the gut microbiome towards more efficient intestinal absorption of calories and increased lipid deposition, which may influence body weight [54]. Moreover, zonulin levels seem to correlate directly with body mass index (BMI) and visceral fat content in obese adults [55].

In addition to changes in the gut microbiome, it is also understood that the low-grade inflammatory status described in patients with obesity may play a role in modulating the intestinal permeability. Specifically, IL-6 has been associated with lower expression of zonulin and claudins in TJs of obese adults [56]. Moreover, higher levels of TNF-α [57] and interleukin-1 beta (IL-1β), which are commonly observed in obese patients, alter the expression of myosin light chain kinase in the enterocytes [58]. Interestingly, obese women who underwent weight loss, following a 4-week very low-calorie diet, reported a reduction in inflammatory markers as well as in endotoxemia, possibly suggesting a reduction in the translocation of bacterial products [59].

## 7. Gut Permeability in Patients with NAFLD

There is evidence suggesting that plasma endotoxin concentrations are increased in paediatric patients with simple steatosis, which suggests the presence of some degree of increased gut permeability already in the initial phases of the disease [60]. Moreover, children diagnosed with NASH show higher levels of LPS concentrations compared to control subjects, hinting at a relationship between bacterial translocation and triggering the immune system in NAFLD [61]. A recent meta-analysis by Luther et al. demonstrated that adult patients with NAFLD were at higher risk for developing a leaky gut compared to healthy controls, with an odd ratio of 5.08 (95% confidence interval: 1.98–13.05) [62]. Of note, both qualitative and quantitative alterations of TJ proteins have been described in patients with NAFLD and with NASH-related cirrhosis [63,64].

Recent evidence has suggested that patients with NAFLD have shown an enhanced gut microbiota proliferation in the small intestine, which has been associated with the progression of liver disease in these patients [9]. Interestingly, the presence of intestinal dysbiosis may also impact upon the expression of TJ, increasing gut permeability and translocation of bacterial products [65]. The composition of gut microbiota may also modulate the abundance of short chain fatty acid (SCFA), which in turn may have a protective effect on the intestinal epithelium, as they promote epithelial cell proliferation and adhesion [66]. Specifically, elevated levels of butyrate induce the release of anti-inflammatory cytokines and promote the integrity of the intestinal barrier [67]. It has also been noted that the intestinal inflammatory milieu could increase the permeability per se and contribute to decompensation events in patients with more advanced liver disease [68].

A recent meta-analysis including studies that investigated the increased permeability in paediatric and adult NAFLD patients, concluded that small intestinal permeability increased with the degree of hepatic steatosis, while no association was found with hepatic inflammation, ballooning or fibrosis [63,69,70]. Conversely, according to Luther et al., patients with histology-proven NASH have increased odds for developing a leakier gut compared to healthy controls, with an odds ratio of 7.21 (95% CI: 2.35–22.13) [62]. From a biological point of view, it is expected that an augmented translocation of bacterial products leads to inflammation and fibrogenesis in the liver, via the stimulation of TLR-4. However, a clear association between gut permeability and fibrosis staging or progression in NAFLD has not been demonstrated so far.

## 8. Manipulation of the Gut Permeability as Potential Therapeutical Target in Patients with NAFLD

Not only excessive food intake, but also specific dietary patterns (i.e., high fructose and high-in-saturated-fat intake) have proved to be strongly associated with alterations of the intestinal barrier [71,72]. As such, tailored dietary advice should be offered to patients with NAFLD. A previous study reported an improvement in visceral adiposity, weight, and serum liver function tests (LFTs) in patients with NAFLD who were following a Mediterranean dietary regime or low-calorie diet for 16 weeks [73]; however, there was no specific change in intestinal permeability [74,75]. Conversely, when obese patients with steatosis underwent a weight-reduction regime and lifestyle modifications for 52 weeks, there was an improvement in gut permeability reaching normal values [76]. Moreover, augmenting the intake of dietary fibres was followed by a reduction in serum zonulin levels, LFTs levels and hepatic steatosis in patients with NAFLD, possibly by altering intestinal permeability [77].

In a recent phase 2a trial, 12 weeks of treatment with lubiprostone induced an improvement in LFTs and hepatic fat fraction in patients with NAFLD. Of note, lubiprostone, a metabolite of prostaglandin E, has been used for treating constipation alone as well as constipation associated with irritable bowel syndrome. The beneficial effect of lubiprostone in patients with NAFLD relies on the improvement in the intestinal motility and permeability, and therefore, in a subsequent reduction in the translocation of bacterial products [78].

Other studies have investigated whether manipulating the intestinal microbiome may improve intestinal permeability in patients with NAFLD. Specifically, allogenic faecal microbiota transplant (FMT) prepared from healthy lean stool donors was able to improve small intestinal permeability, as measured by the lactulose:mannitol test, after 6 weeks in NAFLD patients [79]. Among others, bacteria from the *Bacteroides* spp. have been shown to increase the expression of zonulin and improve epithelial barrier function [80]. A recent study has also suggested that the improvement seen in the intestinal permeability reported after chronic physical exercise is mainly driven by an increase in the *Firmicutes* to *Bacteroides* ratio, as well as to an improvement in microbial diversity [67]. Furthermore, oral administration of *Akkermansia muciniphila* was associated with reduced circulating LPS levels and improved LFTs compared to placebo. Interestingly, *A. muciniphila* was previously shown to up-regulate the expression of some TJ proteins [81] as well as to produce entero-protective SCFA, i.e., acetate and propionate [82]. It has also been argued that the administration of *Faacalibacterium prausnitzii* should be considered to restore the integrity of the intestinal barrier, as this is a major butyrate-producer commensal bacterium [83].

## 9. Conclusions

We have summarised the up-to-date evidence that suggests the presence of increased gut permeability in patients with NAFLD. Whilst leaky gut has been mainly investigated against steatosis and NASH, there is not much evidence with regards to severity of liver disease. Measuring the gut permeability in vivo should become part of the clinical work-up in patients with NAFLD, as this could identify patients who may most benefit from intestinal barrier interventions.

## Figures and Tables

**Figure 1 ijms-23-00662-f001:**
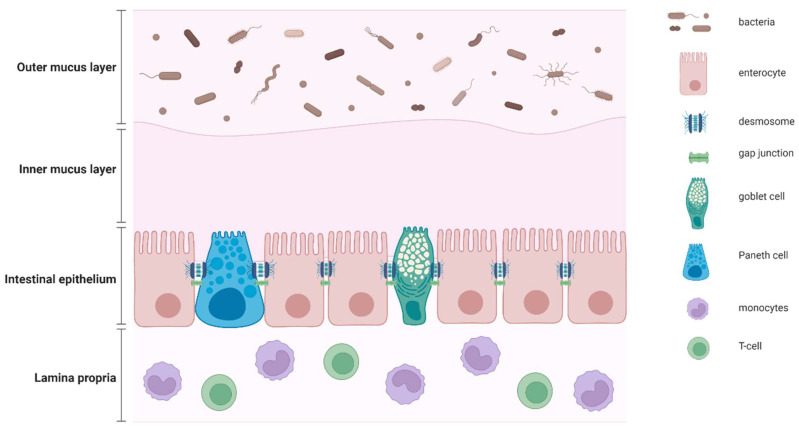
Intestinal mucosal barrier. Epithelial cells, together with Paneth cells and goblet cells, form a layer that acts as a mechanical barrier, which is sealed by a complex combination of desmosomes and gap junctions. The layer of mucus works as a chemical barrier that limits the direct contact between the gut microbiome and the intestinal epithelium. Immune cells are mainly confined to the lamina propria (created with BioRender.com; accessed on 13 December 2021).

**Table 1 ijms-23-00662-t001:** Techniques for in vivo measurement of gut permeability.

Technique	Sample Tested	Advantages	Disadvantages
Orally indigested probes (lactulose to mannitol ratio, sucralose)	Urine	Informative on damage of intestinal mucosa Specific timing of urinary excretion for regional analysis of the intestine Non-invasive	Preparation required (diet/fasting) Time-consuming procedure Not informative on TJs Influenced by kidney function
FABP-2	Serum	Highly specific for small intestine enterocytes Informative on damage of intestinal mucosa Non-invasive and rapid measurement Rapid sample collection	Not informative on TJs False positives (ischemia-reperfusion injury)
Zonulin	Serum	Informative on TJs Non-invasive Rapid sample collection	Lack of standardisation between commercial kits
Calprotectin	Stool	Informative on damage of intestinal mucosa Non-invasive Rapid sample collection	Not informative on TJs False positives (inflammatory bowel disease)
Fluorescence spectroscopy	Transcutaneous assessment	Non-invasive Informative on intestinal permeability and motility Automated analysis	Skin colour and BMI impact on fluorescent measurement Requires specialised equipment Limited evidence in healthy individuals

Abbreviations: TJs: tight junctions, FABP-2: fatty acid binding protein-2, BMI: body mass index.

## Data Availability

All the papers included in this review are quoted among the references.

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
