# Peer review of "The Intestinal Barrier and Its Dysfunction in Patients with Metabolic Diseases and Non-Alcoholic Fatty Liver Disease"

_ijms, 2022, doi:10.3390/ijms23020662_

Round 1

Reviewer 1 Report

Manuscript ID: IJMS-1530178          

Title: THE INTESTINAL BARRIER AND ITS DYSFUNCTION IN PATIENTS WITH NON-ALCOHOLIC FATTY LIVER DISEASE

At first, the authors described the basic conception of the intestinal barrier and gut permeability. Then, the recent evidence concerning the association between gut permeability and various metabolic disorders, including non-alcoholic fatty liver disease (NAFLD), type-2 diabetes mellitus (T2DM), and obesity, were summarized based on previous clinical studies. Although this manuscript is well-written, several critical issues still exist, detailed as follows:

  1. There are a large number of review studies similar to the current topic. Compared to the previous review, the novelty of the current article is absent. Authors should appeal the novelty. Representative similar review articles are shown as follows:
    1) Kessoku, T., Kobayashi, T., Tanaka, K., Yamamoto, A., Takahashi, K., Iwaki, M., Ozaki, A., Kasai,Y., Nogami, A., and Honda, Y. (2021). The Role of Leaky Gut in Nonalcoholic Fatty Liver Disease: A Novel Therapeutic Target. International Journal of Molecular Sciences 22, 8161.
    2) Martín-Mateos, R., and Albillos, A. (2021). The Role of the Gut-Liver Axis in Metabolic Dysfunction-Associated Fatty Liver Disease. Frontiers in Immunology 12, 1265.
    3) Nawrot, M., Peschard, S., Lestavel, S., and Staels, B. (2021). Intestine-liver crosstalk in Type 2 Diabetes and non-alcoholic fatty liver disease. Metabolism 123, 154844.
  2. The authors entitled the current article as “The Intestinal Barrier And Its Dyfunction In Patients With Non-Alcoholic Fatty Liver Disease”. However, the present manuscript summarized the relations between intestinal barrier/permeability and various metabolic diseases, like NAFLD, T2DM, and obesity. Thus, the main title should be changed due to unmatching the contents. In addition, a spelling error is found in the main title. Please change the ‘DYFUNCTION’ into ‘DYSFUNCTION’.
  3. Review papers should give a brief summary of recent trends and research in sepecific filed in order to be easily understood by readers. So, informative tables and/or figures are recommended to be added to the current manuscript.

Reviewer 2 Report

I read with interest mini-review entitled “THE INTESTINAL BARRIER AND ITS DYSFUNCTION IN PATIENTS WITH NON-ALCOHOLIC FATTY LIVER DISEASE”.

I found the topic of this paper very interesting and relevant. The paper is logically presented and nicely written.

Since authors suggest to measure the gut permeability in-vivo as part of the NAFLD work-up, I would suggest to include an additional figure or table with the overview of the different techniques to measure intestinal barrier function, including pros and cons. 

Reviewer 3 Report

This is an interesting review regarding the role of gut permeability in the pathogenesis of NAFLD. The Authors should better stress the role of endotoxin in determining liver damage in NAFLD. In addition they should discuss the emerging role of oxidant stress and its relationship with endotoxin in this clinical setting.

Reviewer 4 Report

The paper “THE INTESTINAL BARRIER AND ITS DYSFUNCTION IN PATIENTS WITH NON-ALCOHOLIC FATTY LIVER DISEASE" by Forlano et al. is a review which summarize the association between intestinal barrier and NAFLD.

The article is well written. The study has a good design. The article is logically divided into sections and subsections. There is one figure of good quality. The references cited are relevant and adequate. The work has an average degree of novelty and of good interest to the readers.

Comments:

  • Background: it should be explained the reason underlying the association between intestinal barrier and NAFLD development. In fact, over 50% of the blood supply in the liver is provided by the splanchnic district, thus being among the organs most exposed to toxins of intestinal origin, as it represents the first line of defence against products of bacterial origin (DOI: https://doi.org/10.3390/pr9010135).
  • Another important issue in NAFLD patients is that together with a dysfunctional intestinal barrier, we assist to a gut microbiota proliferation in the small intestine. The association of these two events can further enhance the passage of bacterial metabolites in the blood stream, with increased risk of NAFLD development and progression (https://doi.org/10.3390/pr9010135; 3390/antiox10020270)

Round 2

Reviewer 1 Report

  1. We don't find Table 1 in the main text. please add it.
  2. In the title, please change "metabolic disease" into "metabolic diseases".
  3. please revise the new additional references into IJMS format.

Author Response

Thank you very much for the comments. We have amended the manuscript accordingly